# Antioxidant Capacity Changes and Untargeted Metabolite Profile of Broccoli during Lactic Acid Bacteria Fermentation

**Feixiang Hou [1,2], Yanxue Cai [1,\*] and Jihui Wang [1,2,\*]**

1    Key Laboratory of Healthy Food Development and Nutrition Regulation of China National Light Industry, School of Life and Health Technology, Dongguan University of Technology, Dongguan 523808, China
2    School of Bioengineering, Dalian Polytechnic University, Dalian 116034, China
\*    Correspondence: caiyanxue@dgut.edu.cn (Y.C.); wangjihui@dgut.edu.cn (J.W.)

**Abstract:** The purpose of this study was to reveal the changes in total phenolic content and antioxidant capacity of broccoli, and an untargeted metabolomics approach was developed to investigate the effect of lactic acid bacteria fermentation on the metabolome of broccoli florets. The results showed that the total phenolic content and antioxidant capacity significantly increased after fermentation. The untargeted metabolite profile showed that the main chemical components of fermented and unfermented broccoli are lipids and lipid-like molecules, organic acids and derivatives and organoheterocyclic compounds. Univariate and multivariate statistical analyses of the identified metabolites showed some metabolites such as sorbitol are upregulated after fermentation, and that other metabolites such as l-malic acid are downregulated after fermentation. Moreover, metabolite pathway analyses were used to study the identification of subtle but significant changes among groups of related metabolites that cannot be observed with conventional approaches. KEGG pathway analysis showed that metabolites are mainly enriched in the glucagon signaling pathway, pyruvate metabolism, glycolysis/gluconeogenesis and fructose and mannose metabolism after fermentation, compared with raw broccoli. The results of this study can help to further our understanding of the impact of LAB fermentation on bioactivity changes in and the metabolites profile of fermented broccoli, and the application of fermented broccoli in health foods and special dietary foods.

**Keywords:** fermented broccoli; lactic acid bacteria; antioxidant capacity; untargeted metabolite profile





## 1. Introduction

The consumption of essential nutrients in vegetables is crucial for the promotion and maintenance of human health. Among them, broccoli (*Brassica oleracea* var. *italica*) is a good source of health-promoting compounds such as phenolic compounds, vitamins and glucosinolates [1]. In addition, epidemiological studies have shown that regular intake of broccoli may reduce the risk of many chronic diseases such as cancer and type II diabetes [2].

However, fresh vegetables have a short shelf-life and are easily contaminated by microorganisms. Some processing methods such as cooking will cause changes in their phytochemical properties and reduce their nutritional value [3]. Lactic acid bacteria (LAB) fermentation has been widely used as a mild processing approach for improving the nutritional, medicinal and hygienic properties of fruits and vegetables. Vegetables are considered good media for the growth of lactic acid bacteria (LAB) due to their high content of bioactive compounds such as vitamins and antioxidant compounds [4–6]. Hydrolases produced by LAB during fermentation can release bound phenols by breaking down the cell wall, thereby increasing the content of phenolic substances. LAB have the effect of decomposing free esters and bound phenolic acids in fruit and vegetables into free phenolic acids, and enhancing their biological activity. Moreover, LAB can protect certain polyphenols from chemical degradation under physiological conditions. The effects of LAB fermentation on the phytochemical and functional properties of broccoli have also been

reported [7,8]. However, the mechanisms of the metabolites' changes, which are caused by fermentation, are not clear.

The effect of different treatments on the metabolome of broccoli sprouts was studied by Ming Tian et al. and Mariateresa Maldini et al. using untargeted metabolomics [9,10]. Yuge Guan et al. [11] used untargeted metabolomics and physiological analyses to validate the molecular mechanism of the response to wounding stress in broccoli florets and shreds; they observed that wounding stress activated glucosinolate and phenylpropanoid biosynthesis by regulating the levels of the precursors, including L-leucine, phenylalanine, tyrosine, valine, isoleucine, tryptophan, methionine, and phenylalanine. There is a paucity of information about changes in the untargeted metabolite profile of broccoli after LAB fermentation.

Metabolomics, as an important component of omics [12], can amplify small and imperceptible changes in the expression of genes and proteins through metabolites. Since metabolomics can discover subtle differences between samples, it is increasingly being applied in the field of food processing to study nutritional and bioactivity changes in processed foods, caused by multiple factors.

Untargeted metabolomics methods filter the entire metabolite content of samples to characterize the molecular phenotype and compare metabolite profiles among different sample groups [13]. Recently, high-throughput metabolomics methods have focused on providing an inclusive metabolite profile of biological samples [14]. GC-MS-based metabolomics offers an overview of the pathways and is only suited for compound classes appearing mainly in primary metabolism. For instance, novel UPLC-TOF-MS-based metabolomics provides a platform for estimating at a very low level, with better accuracy, sensitivity, precision, and time-effectiveness [15]. Namgung et al. used metabolomics to study the dynamic changes in metabolites during the fermentation of Korean soybean paste (Doenjang) [16]. Lee et al. combined 1H nuclear magnetic resonance (NMR) and GC-MS-based metabolomics to explore the transformation of the metabolite profile of *Oenococcus oeni* in wine during fermentation [17]. Tomita et al. identified the species dependence and characteristics of *Lactobacillus* fermentation using NMR-based metabolomics [18].

A large number of LAB exist in fermented vegetables, and some strains with the ability to metabolize polyphenols have been screened; however, the specific mechanism of transformation is not completely clear, and it may be explored in depth in order to convert polyphenolic substances into substances with higher bioavailability and bioactivity through fermentation. Metabolomics could help us to figure out the different expression trends of each metabolite during fermentation, which may then help us to understand the relevant pathways involved in the changing biological activity of broccoli before and after fermentation. Therefore, in this study, two LAB strains isolated from broccoli florets were used to ferment broccoli. Subsequently, the changes in the total phenolic content and antioxidant capacity, based on 2,2-diphenyl-1-picrylhydrazyl (DPPH), 2,2′-azino-bis(3-ethylbenzothiazoline-6-sulfonic acid) (ABTS) and ferric-reducing antioxidant power (FRAP) antioxidant methods of broccoli during LAB fermentation, were studied. The untargeted metabolite profile of the selected broccoli samples following LAB fermentation was obtained using UPLC-TOF-MS. In addition, univariate and multivariate statistical analyses were performed to characterize the metabolites of broccoli during LAB fermentation. This study aimed to validate the mechanism of response to LAB fermentation in broccoli; the comprehensive and dynamic metabolite changes were explored using metabolomics and physiological analyses. Furthermore, the Kyoto Encyclopedia of Genes and Genomes (KEGG) pathway analysis was applied to analyze the important metabolites and critical pathways induced by LAB fermentation in broccoli. This study provides a comprehensive approach to assessing bioactive metabolites in broccoli.

## 2. Materials and Methods

### 2.1. Materials

Broccoli (*Brassica oleracea* var. *italica*) was purchased from the supermarket in Daling-shan Town (Dongguan, China). MRS (de Man, Rogosa and Sharpe) broth and MRS agar were purchased from Huankai Microbial (Guangdong, China). Folin–Ciocalteu reagent was purchased from Solarbio (Beijing, China). ABTS was purchased from Macklin Biochemical (Shanghai, China). DPPH and 2,4,6-tri(2-pyridyl)-1,3,5-triazine (TPTZ) were purchased from Aladdin Biochemical (Shanghai, China). Glucono delta-lactone (GDL) was purchased from Yuanye Bio-Technology (Shanghai, China). Acetonitrile was purchased from Merck (Shanghai, China). Ammonium acetate was purchased from Merck (Shanghai, China). All the other chemicals were of analytical grade or higher.

### 2.2. Isolation and Identification of LAB

LAB were isolated from broccoli florets that were obtained from a local supermarket (Dongguan, China), using a previous method [19]. Shredded broccoli florets (100 g) were added to 250 mL MRS broth medium and incubated at 37 °C for 24 h. After incubation, an appropriate concentration was selected, and 200 μL was transferred to MRS agar medium. Single colonies on the surface of the plate were selected and streaked onto MRS agar medium according to Gram staining, and microscopic examination was carried out for preliminary identification. Ten isolated strains were selected and stored at −20 °C in MRS broth containing 25% sterile glycerol. The LAB strains were identified at Guangdong Magigene Biotechnology Co., Ltd. (Guangzhou, China). Genomic DNA was extracted and purified. The 16S rDNA of isolates was amplified by PCR using the primer 8F and 16SR. The 16S rDNA gene sequences were determined using an ABI3730 sequencer (Applied Biosystems, Waltham, MA, USA). Multiple sequences were assembled and compared with type strain sequences using BLAST (Basic Local Alignment Search Tool) with a nucleotide database on NCBI (http://www.ncbi.nlm.nih.gov/blast/Blast.cgi, accessed on 28 July 2022), and the taxonomy of the strains was obtained. *Lactiplantibacillus plantarum* (LAB1) and *Lactiplantibacillus pentosus* (LAB10) were identified and were chosen for fermentation.

### 2.3. Preparation of Broccoli Puree

Broccoli florets were cut at approximately 2 cm below the crown. 400 mL of sterilized deionized water was added to the kitchen scale blender per 600 g shredded broccoli florets, and homogenized using the wall break mode for 75 s. After mixing well, the broccoli puree was aliquoted into a sterile media bottle (250 g).

### 2.4. Fermentation

LAB strains (LAB1, LAB10) which were stored at −20 °C were inoculated into MRS broth medium at 37 °C for 24 h. After cultivation, 1 mL was inoculated into another MRS broth medium at 37 °C for 24 h. The cultures were centrifuged at $4000 \times g$ for 10 min at 4 °C, washed twice with sterile phosphate-buffered saline (PBS, pH = 7.4) and dissolved in sterile deionized water. Then, the LAB1 and LAB10 were mixed and added to the broccoli puree to obtain a concentration of 8 log CFU/g broccoli puree. The fermentation was performed at 37 °C. The broccoli puree was fermented for 0, 12, 24, 36, 48 and 60 h. Compared with the raw broccoli puree, 0 h of fermentation time meant the LAB strains (LAB1, LAB10) were added to the broccoli puree without fermentation; after the LAB strains were added, the samples were immediately stored. After the fermentation was completed, the ferments were randomly divided into three samples and stored at 4, −20, and −80 °C.

### 2.5. Total Phenolics Content

The total phenolics content was determined using the Folin–Ciocalteu method [19]. Briefly, the supernatant (20 μL) of a sample (5 mg/mL) was mixed with 180 μL deionized water, 800 μL of sodium carbonate solution (7.5 g/L) and 1000 μL of 0.2 N Folin–Ciocalteu reagent. After 1 h of incubation in the dark at 37 °C, the absorbance was measured at 765 nm.

The reaction was performed using a sample or gallic acid standard (0.04 to 0.18 mg/mL). The results were expressed as gallic acid equivalent (mg gallic acid/g DW).

### 2.6. ABTS Radical Scavenging Activity

The radical scavenging activity of ABTS was evaluated using a previous method with slight modifications [20]. Equivalent volumes of ABTS solution (7 mmol/L) and $K_2S_2O_8$ (2.45 mmol/L) solution were mixed in the dark for 16 h to produce ABTS radical cations. Then, the mixture was diluted with PBS to obtain an absorbance of $0.70 \pm 0.02$ at 734 nm. The supernatant (400 μL) of the sample (2 mg/mL) was collected and mixed with 3600 μL ABTS working solution in the dark for a 6 min reaction, and the absorbance was measured at 734 nm. The reaction was performed using a sample or Trolox standard (0.15 to 0.35 mmol/L). The results were expressed as a radical scavenging activity (%) and Trolox equivalents (mmol Trolox/L). Scavenging activity was calculated using the following formula:

$$\text{ABTS RSA}(\%) = (A_{\text{blank}} - A_{\text{sample}})/A_{\text{blank}} \times 100\%. \tag{1}$$

### 2.7. DPPH Radical Scavenging Activity

DPPH's radical scavenging activity was evaluated using a previous method with slight modifications [21]. The supernatant (250 μL) of the sample (5 mg/mL) was collected and mixed with 500 μL DPPH solution (45 mg/L), which was dissolved in 80% methanol. Then, the mixture was kept in the dark for 30 min and absorbance was measured at 517 nm. The reaction was performed using a sample or Trolox standard (0.02 to 0.08 mmol/L). The results were expressed as radical scavenging activity (%) and Trolox equivalents (mmol Trolox/L). Scavenging activity was calculated using the following formula:

$$\text{DPPH RSA}(\%) = (A_{\text{blank}} - A_{\text{sample}})/A_{\text{blank}} \times 100\%, \tag{2}$$

### 2.8. Ferric-Reducing Antioxidant Power

FRAP was determined using a previous method with slight modifications [20]. The reaction mixture was prepared by mixing 300 mL of acetate buffer (0.1 mol/L, pH = 3.6), 30 mL of TPTZ solution (10 mmol/L), and 30 mL of $FeCl_3$ solution (20 mmol/L). The supernatant (100 μL) of the sample (2 mg/mL) was mixed with 3000 μL of the reaction mixture at 37 °C for 30 min, and the absorbance was measured at 593 nm. The reaction was performed using a sample or Trolox standard (0.05 to 0.25 mmol/L). The results were expressed as Trolox equivalents (mmol Trolox/L).

### 2.9. Untargeted Metabolomics

Precooled methanol/acetonitrile/water solution (2/2/1, *v/v*) was added to an appropriate amount of sample stored at −80 °C and vortexed. Then, the mixture was sonicated at 4 °C for 30 min, placed at −20 °C for 10 min and centrifuged at 14,000× *g* at 4 °C for 20 min. The supernatant was taken for vacuum drying. Then, 100 μL of acetonitrile solution (acetonitrile/water = 1/1, *v/v*) was added to the vacuum-dried sample and vortexed. The mixture was centrifuged at 14,000× *g* at 4 °C for 15 min, and the supernatant was used for mass spectrometry. Chromatographic separation was performed using an Agilent 1290 Infinity LC composed of an ACQUITY UPLC BEH C-18 column (1.7 μm, 2.1 mm × 100 mm) combined with an AB Triple TOF 6000. The mobile phase A was 25 mmol/L ammonium acetate and 0.5% formic acid in water. The mobile phase B was methanol. The column temperature was maintained at 40 °C. The flow rate was 0.4 mL/min and the injection volume was 2 μL. The gradient elution was as follows: 0–0.5 min, 5% B; 0.5–10 min, 5–100% B; 10–12 min, 100% B; 12.0–12.1 min, 100–5% B; 12.1–16 min, 5% B. The mass spectrometry parameters were as follows: ion source gas 1 and gas 2, 60; curtain gas, 30; Source temperature, 600 °C; ion spray voltage floating, ±5500 V; TOF MS scan *m/z* range, 60–1000 Da; product ion scan *m/z* range, 25–1000 Da; TOF MS scan accumulation time, 0.20 s/spectrum;

product ion scan accumulation time, 0.05 s/spectrum. The secondary mass spectrometry was acquired using information-dependent acquisition (IDA) with a high sensitivity mode; the declustering potential was $\pm 60$ V, and the collision energy was $35 \pm 15$ eV. The IDA settings were as follows: exclude isotopes within 4 Da, candidate ions to monitor per cycle.

### 2.10. Metabolite Annotation and KEGG Pathway Enrichment Analysis

An in-house database (Shanghai Applied Protein Technology) was used to annotate metabolites by matching molecular mass data ($m/z$) (mass error < 10 ppm), MS/MS spectrometric fragments and the retention time of the samples. The online KEGG (Kyoto Encyclopedia of Genes and Genomes) database (http://www.kegg.jp/kegg/pathway.html, accessed on 28 July 2022) was also used to annotate the metabolites and to perform metabolic pathway enrichment analysis of significantly differential metabolites by integrating the metabolites into pathways. A Fisher's exact test was conducted for calculating the significance of enrichment in each pathway.

### 2.11. Statistical Analysis

All the experiments were performed in triplicate, and the results were analyzed using IBM SPSS Statistics 27.0 (IBM Corp., Armonk, NY, USA) with a one-way ANOVA. An LSD test was applied for the comparison of mean values, and a Waller–Duncan test for the identification of significant differences ($p < 0.05$). The data were plotted using OriginPro 2023 (OriginLab, Northampton, MA, USA).

## 3. Results

### 3.1. Total Phenolic Content

A gallic acid standard curve was generated ($A = 0.32172c + 0.04152$, $R^2 = 0.9987$) and values were expressed as gallic acid (GA) equivalents (mg GA/g DW), as shown in Figure 1. Raw broccoli's total phenolic content was $12.3 \pm 0.7$ mg GA/g DW, which is in the same range as a previous report [22]. After fermentation, the total phenolic content (TPC) values of the fermented broccoli powder increased significantly ($p < 0.05$). The content of total phenols showed an increasing trend with the increase in fermentation time. After 48 h of fermentation, the increase in TPC slowed down and no significant difference was observed compared to the results of 60 h. The TPC reached the highest amount of $31.7 \pm 0.4$ mg GA/g DW after 60 h of fermentation time, which is an increase of ~157% compared with the raw broccoli. A similar trend in TPC following LAB fermentation has been reported in other studies [7,19]. The increase in TPC could be caused by various factors such as the release of bound polyphenols or the conversion of complex polyphenols to smaller phenols. Production of cell wall polysaccharide-degrading enzymes by LAB, such as cellulase and pectinases, could also cause the release of bound polyphenols [23]. LAB also produce a series of enzymes, such as glycosidases, tannases and esterases, that convert phenolic acid esters into aglycones and phenolic acids with higher antioxidant capacity [24,25], thereby increasing TPC. Furthermore, the dissociation of protein–polyphenol complexes caused by microbial proteases could also result in higher TPC [26]. Essentially, it must be noted that the Folin–Ciocalteu method for the determination of TPC is not a direct method of measuring TPC, but instead an antioxidant capacity measurement of the ability of electron-donating compounds [27]. Because the stability of phenolic compounds is pH-dependent, the lower pH also contributed to higher TPC [28].

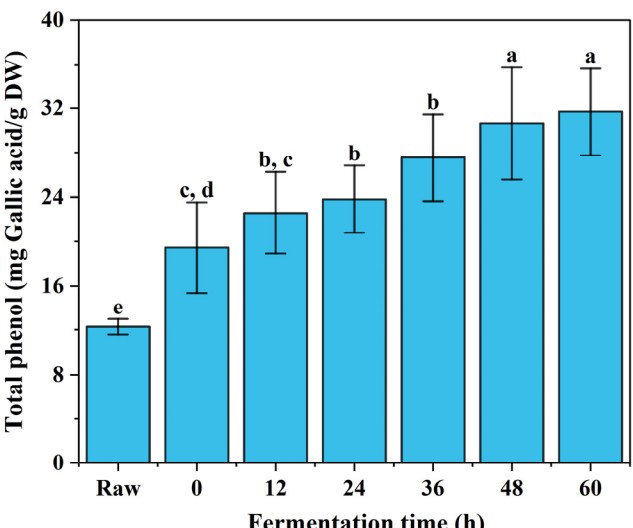

**Figure 1.** The total phenolic content changes of freeze-dried fermented broccoli powder. (Different letters indicate significantly different, *p* < 0.5).

### 3.2. Antioxidant Activity

It has been confirmed that phenolic compounds act as reducing agents, free radical scavengers and singlet oxygen quenchers, with their antioxidant capacity mainly attributed to the hydrogen atom transfer or electron donation to free radicals [29]. Therefore, the antioxidant capacity of fermented broccoli was analyzed based on DPPH, ABTS and FRAP methods, and the results are presented in Figure 2. Raw broccoli's Trolox equivalent antioxidant capacities based on DPPH, ABTS and FRAP methods were $0.050 \pm 0.006$, $0.174 \pm 0.017$ and $0.069 \pm 0.004$ mmol Trolox/L, all of which are in same range as a previous report [30]. After fermentation, DPPH and ABTS's radical scavenging capacity and FRAP were significantly ($p < 0.05$) increased. The highest DPPH radical scavenging capacity was observed in fermented broccoli after 60 h of fermentation, which is within an increase of ~22% compared with the raw broccoli. Broccoli samples after 30 h of fermentation exhibited the highest ABTS scavenging capacity, within an increase of ~74% compared with the raw broccoli. In the case of FRAP, 12 h of fermentation time exhibited the highest FRAP, with a ~79% increase compared with the raw broccoli. The same trend was reported on antioxidant activity changes in other fermented broccoli products [8]. As shown in Figure 2, the DPPH and ABTS radical scavenging capacity showed a similar trend during LAB fermentation, but a greater increase was observed in the case of ABTS methods. This may be due to the different antioxidant mechanisms of DPPH and ABTS radical scavenging; the DPPH method is based on electron transfer, whereas the ABTS method is based on hydrogen atom transfer [31]. Furthermore, the result of DPPH's radical scavenging capacity suggested that LAB fermentation might improve the availability of polyphenol compounds with proton-donating properties [32], because a similar trend is seen between the result of DPPH radical scavenging capacity and the TPC, and the mechanism of the increase in TPC is also based on electron transfer. In addition, a similar change was found in FRAP, which may be attributed to the influence of LAB fermentation on the antioxidant capacities of phenolic and polyphenolic compounds [30]. Several studies have reported a similar trend in antioxidant capacity following LAB fermentation of broccoli [19] and other products [33,34].

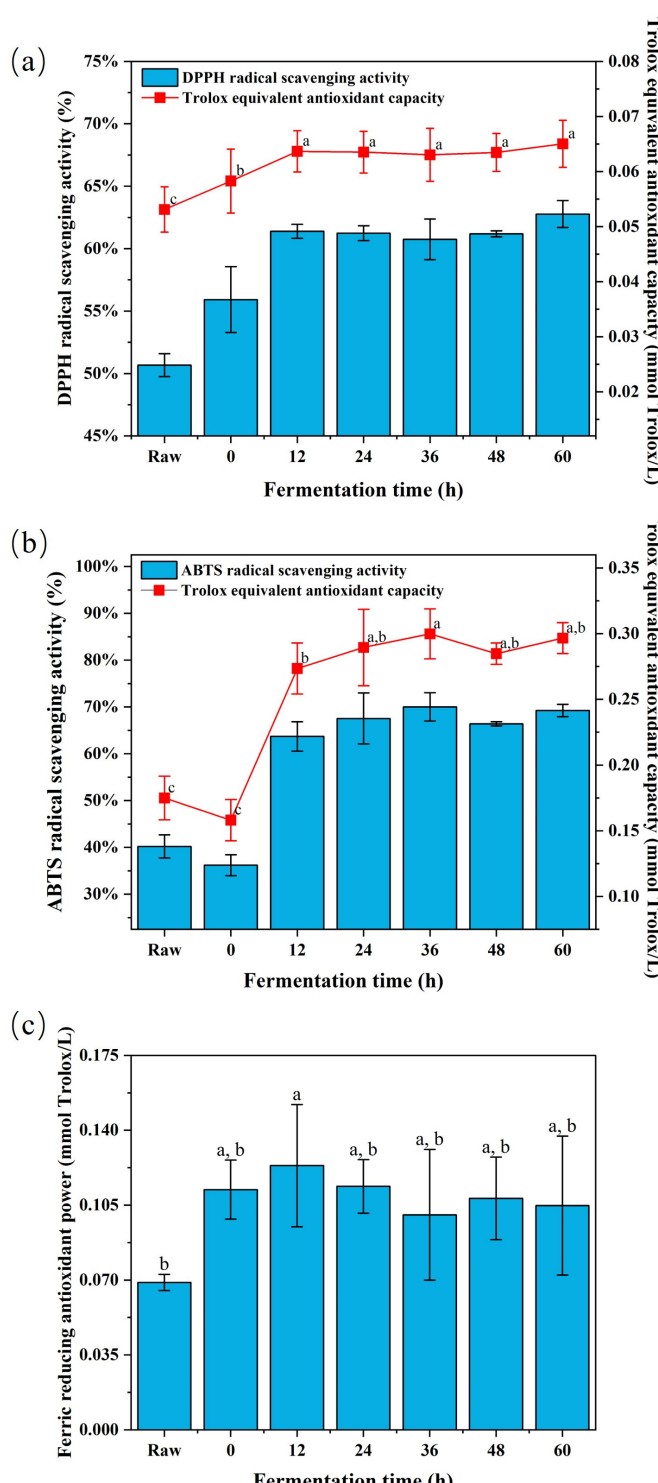

**Figure 2.** (**a**) DPPH radical scavenging activity, (**b**) ABTS radical scavenging activity, and (**c**) ferric-reducing antioxidant power of freeze-dried fermented broccoli powder. (Different letters indicate significantly different, *p* < 0.5).

### 3.3. Untargeted Metabolite Profiling

For the untargeted metabolite profile analysis, we only selected the two most representative fermentation time points, 24 h and 48 h, based on the significant variability in their TPC contents, as shown above. GDL (glucono delta-lactone) is a generally recognized as safe (GRAS) substance; it is a weak acid which has been widely used in acidified food products to improve product texture [35]. Therefore, acidification was performed on raw

broccoli puree using GDL to achieve pH = 4 broccoli puree for assessing the influence of acidification without fermentation. In total, Four samples were selected for untargeted metabolite profile analysis. A total of 977 metabolites (589 in negative ion mode and 388 in positive ion mode) were putatively identified in all four broccoli samples, as shown in Table S1. The primary chemical classification of 977 metabolites was performed based on chemical taxonomy. The quantitative proportions of metabolites are shown in Figure 3. The top three chemical classes were lipids and lipid-like molecules, organic acids and derivatives and organoheterocyclic compounds. A previous study also showed that lipids and lipid-like molecules and organic acids and derivatives were the primary chemical compositions in LAB-fermented broccoli [27].

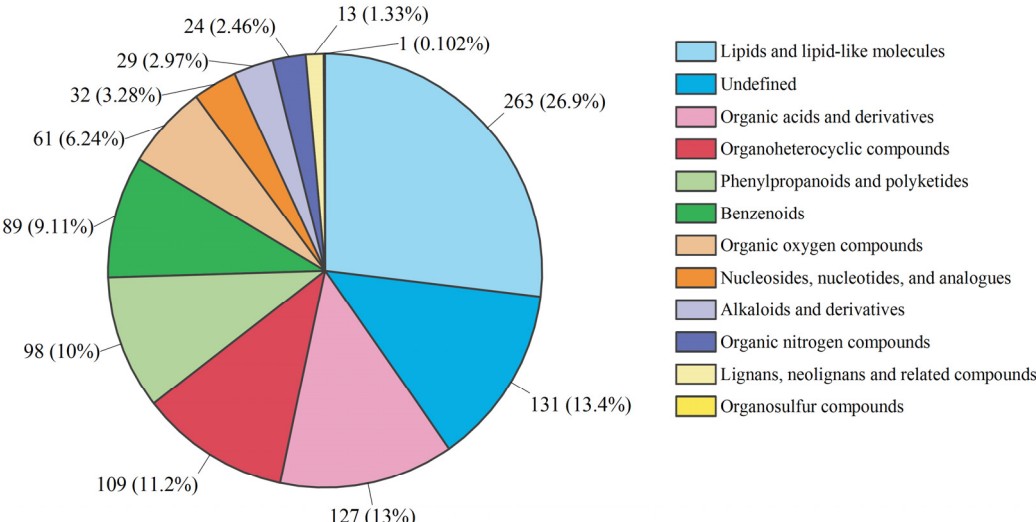

**Figure 3.** Quantitative proportion of identified metabolites in each chemical class. Different color blocks represent different chemical classifications, and the percentage represents the percentage of the number of metabolites in the chemical classification attributed to all metabolites. Metabolites without chemical class assignment were classified as undefined.

Univariate statistical analyses were performed for all identified metabolites. Fold change (FC) analysis was performed on all metabolites including the undefined chemical class. Differential metabolites with FC > 1.5 or FC < 0.67 and a $p$-value < 0.05 are highlighted. Volcano plots are presented in Figure 4. Compared with the raw broccoli, there were 164 downregulated metabolites (69 in negative mode and 95 in positive mode) and 129 upregulated metabolites (47 in negative mode and 82 in positive mode) in the 24 h, and there were 235 downregulated metabolites (112 in negative mode and 123 in positive mode) and 182 upregulated metabolites (68 in negative mode and 114 in positive mode) in the GDL. The top five upregulated metabolites of the raw broccoli vs. the 24 h comparison in negative mode were l-malic acid, guanosine, cyclic guanosine monophosphate, *N,N*-dimethyl guanosine, and panasenoside, while penciclovir, [3-hydroxy-1-(4-methoxy-7-oxofuro[3,2-g]chromen-9-yl)oxy-3-methylbutan-2-yl] (*E*)-2-methylbut-2-enoate, atazanavir, flumequine, and (+)-palmitoylcarnitine were in positive mode. In contrast, sorbitol, calcium pantothenate, l-(−)-3-phenyllactic acid and two more undefined compounds in negative mode, and [4a,7-dihydroxy-7-methyl-1-[3,4,5-trihydroxy-6-(hydroxymethyl)oxan-2-yl]oxy-1,5,6,7a-tetrahydrocyclopenta[c]pyran-5-yl] (*E*)-3-(4-methoxyphenyl)prop-2-enoate, diosmetin-7-*O*-rutinoside, ritonavir, germinaline, and homoharringtonine in positive mode were the top five downregulated metabolites after 24 h of fermentation time, compared with the raw broccoli.

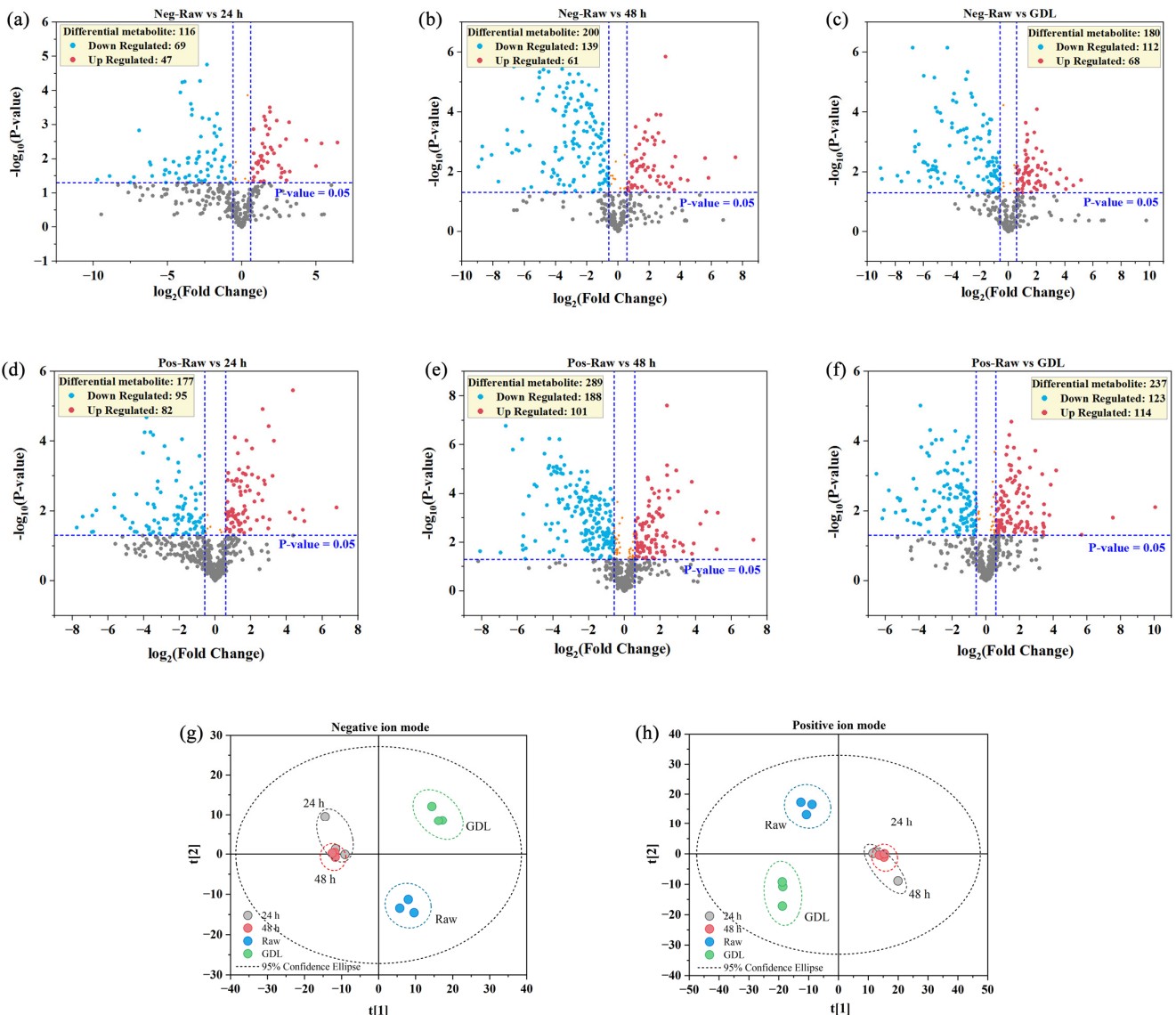

**Figure 4.** Volcano plots of identified metabolites. (**a–c**) are negative ion mode and (**d–f**) are positive ion mode. The points in red are upregulated differential metabolites, and the points in blue are downregulated. Orange points indicate significant expression but less difference. Grey points mean insignificant expression and less differences. A principal component analysis 2-D score plot of (**g**) negative ion mode and (**h**) positive ion mode.

Multivariate statistical analysis was performed. The principal component analysis (PCA) 2-D score plots are presented in Figure 4g,h. The first and second principal components explained 42.5% and 21.1% of the variations and 42.5% and 20.5% of the variations in the negative and positive ion modes, respectively. Figure 4 shows that the raw broccoli, the samples after fermentation (24 and 48 h), and the GDL deviated from each other and were clustered. Combined with the results of the volcano plots in the univariate statistical analyses, the PCA indicated that the raw broccoli, the 24 and 48 h, and the GDL were independent and distinct from the others, which shows that overall, the metabolite profile was affected by LAB fermentation and acidification by GDL.

### 3.4. OPLS-DA Analysis

An orthogonal partial least squares-discriminant analysis (OPLS-DA) was conducted for six comparisons among four groups in each mode. Each OPLS-DA model passed the permutation test and $Q^2 > 5$, indicating that the models were stable and reliable. Differen-

tial metabolites with OPLS-DA VIP > 1 and a *p*-value < 0.05 were selected as significant differential metabolites and shown in Figure S1. Among all the upregulated metabolites, the most commonly occurring (and those that occurred more than once) were sorbitol, L-lactic acid, kaempferol, l-(−)-3-phenyllactic acid, 3-(4-hydroxy-3-methoxyphenyl)propionic acid, alanyl phenylalanine, tryptophan, gluonic acid, gluconic acid, (2-hydroxy-3-octadeca-9,12,15-trienoyloxypropyl)-2-(trimethylazaniumyl)ethyl phosphate, ethylparaben, thymine, quinolactacin A, Ala-Leu and 3,4-dihydroxybenzaldehyde. Among the downregulated metabolites were l-malic acid, guanosine, panasenoside, gluonic acid, gluconic acid, quinic acid, penciclovir, nicotinamide, bonactin and d-glucosamine. Sorbitol has been found to improve the shelf-life quality of fruits [36], and was found upregulated after fermentation compared with the raw broccoli. L-Lactic acid is one of the main products of the growth of LAB. Kaempferol was found to have antioxidant and anti-inflammatory effects [37]. l-Malic acid is one of the main materials for the growth of LAB. Quinic acid is one of the phenolic compounds in broccoli [7], and a previous study also observed a decrease in it after fermentation [38].

*3.5. KEGG Pathway Analysis*

Based on the KEGG pathway analysis, significant differential metabolites were merged into metabolic pathways. KEGG enrichment analysis was performed on six comparisons and shown in Figure 5. Compared with the raw broccoli, the top three KEGG pathways for the metabolites in fermented broccoli to be enriched were the glucagon signaling pathway, pyruvate metabolism, and glycolysis/gluconeogenesis and fructose and mannose metabolism. Compared with the raw broccoli, the top three KEGG pathways for the metabolites in GDL were glyoxylate and dicarboxylate metabolism, pyrimidine metabolism and proximal tubule bicarbonate reclamation. In the case of comparison between the raw and fermented broccoli puree (24 h and 48 h), three of the four enriched KEGG pathways were those of carbohydrate metabolism, which is consistent with the conclusion that lipids and lipid-like molecules accounted for the majority of the chemical composition of the identified metabolites. On the other hand, compared with the raw broccoli, pyrimidine metabolism enriched the most abundant and significantly differential metabolites in fermented broccoli puree, followed by the longevity regulating pathway and worm and HIF-1 signaling pathway.

Existing studies have shown that the precursors of plant polyphenols are derived from the intermediate products of sugar metabolism [39] and synthesized by the shikimic acid pathway [40,41] and flavonoid metabolism pathway. Among them, GA is synthesized from an intermediate product of the shikimic acid pathway [42], 3-hydroxyshikimic acid, under the action of related enzymes, and the remaining shikimic acid pathway plays a central role in the secondary metabolic pathway of plants, providing substrates for other secondary metabolic pathways [43]. The carbon metabolism pathway mainly enriched in this study also proves that some of the sugar metabolism products mainly produced by LAB in the fermentation process of broccoli have a good potential to be converted into phenolic compounds, which may also be one of the reasons for the improvement of the antioxidant capacity of broccoli after fermentation.

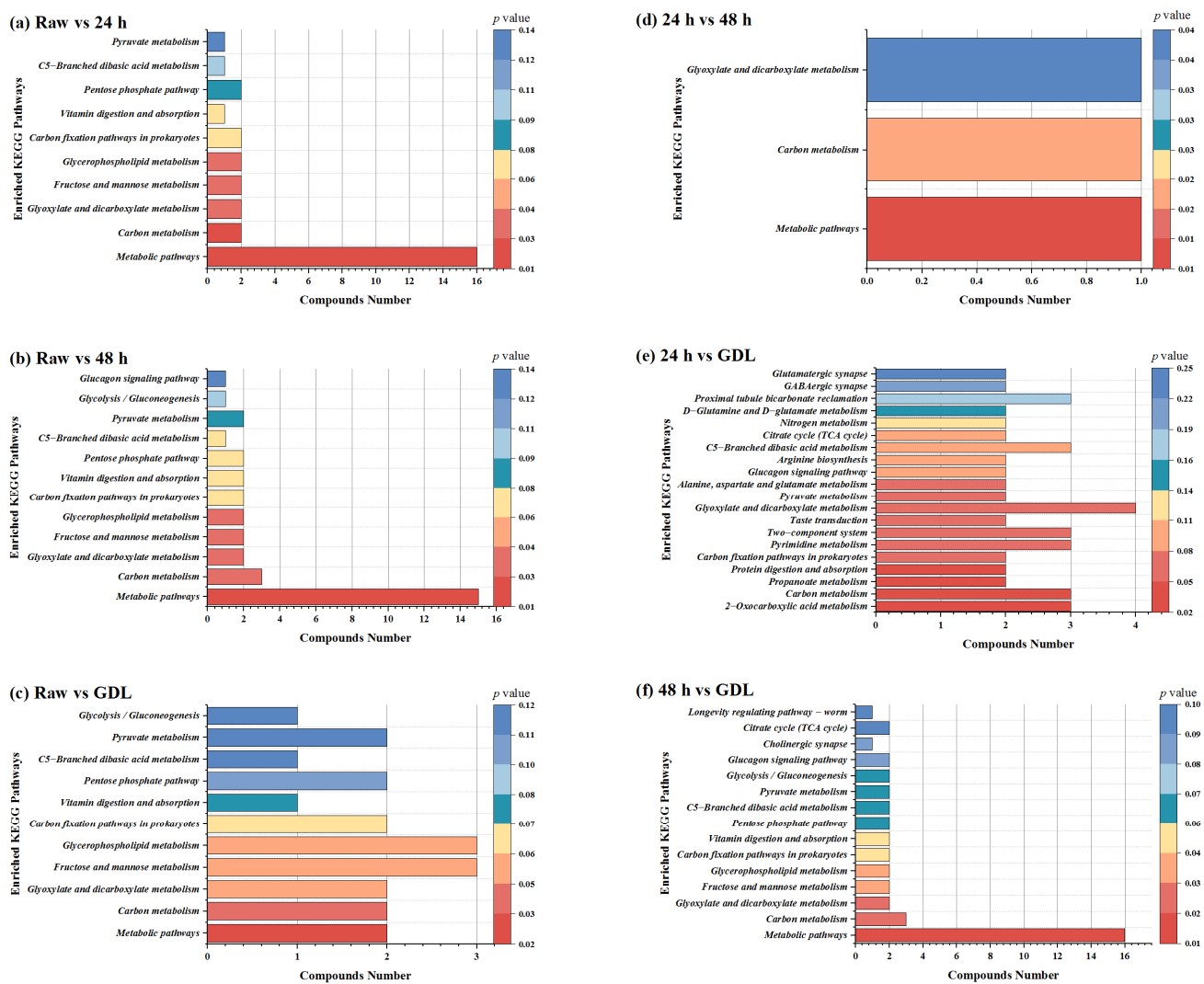

**Figure 5.** KEGG enrichment analysis of metabolic pathways. (**a**–**f**) are six comparisons among four groups of samples between Raw, 24 h, 48 h and GDL groups, respectively.

## 4. Conclusions

In this study, we investigated and reported the variations in the total phenolics and antioxidant capacity between raw broccoli and broccoli following LAB fermentation, and also reported on the untargeted metabolite profile of raw broccoli, fermented broccoli and broccoli acidified with GDL without fermentation. Our results showed that the broccoli puree following LAB fermentation reached the highest TPC, showing a ~157% increase after 60 h of fermentation time compared with the raw broccoli. Antioxidant capacity based on DPPH, ABTS and FRAP methods of fermented broccoli showed a ~22%, ~74% and ~79% increase after 60, 36 and 12 h of fermentation time, respectively. The untargeted metabolite profile showed that the main chemical components of fermented and unfermented broccoli are lipids and lipid-like molecules, organic acids and derivatives and organoheterocyclic compounds. A total of 977 metabolites (589 in negative ion mode and 388 in positive ion mode) were putatively identified in the untargeted metabolite profile. In the case of fermented broccoli and acidification by GDL, univariate and multivariate statistical analyses showed 293 metabolites are upregulated after fermentation, and 417 metabolites are downregulated after fermentation. KEGG enrichment analysis showed that metabolites are mainly enriched in the glucagon signaling pathway, pyruvate metabolism, glycolysis/gluconeogenesis and fructose and mannose metabolism after fermentation, compared with the raw broccoli. Overall, the present study provides a method of revealing the

changes in the antioxidant capacity of broccoli puree during LAB fermentation. Our evaluation of LAB fermentation with regard to metabolites and metabolic pathways should also provide important information for the isolation of key functional compounds, and for the improvement of fermented broccoli in both flavor and application to support a healthy daily diet.

**Supplementary Materials:** The following supporting information can be downloaded at: https://www.mdpi.com/article/10.3390/fermentation9050474/s1, Figure S1: Visualization of significant differential metabolites. Six comparisons among four groups of samples in each mode. (a–c) are negative ion mode, and (d–f) are positive ion mode. Blocks in orange are upregulated metabolites, and blocks in blue are downregulated; Table S1: The list of a total of 977 identified metabolites in fermented broccoli samples.

**Author Contributions:** Investigation, writing—original draft preparation, F.H.; writing—review and editing, funding acquisition, Y.C.; writing—review and editing, supervision, J.W. All authors have read and agreed to the published version of the manuscript.

**Funding:** This research was funded by the National Natural Science Foundation of China, grant number 32001662 and the Foundation for Innovation Team in Higher Education of Guangdong Province, grant number 2021KCXTD035.

**Institutional Review Board Statement:** Not applicable.

**Informed Consent Statement:** Not applicable.

**Data Availability Statement:** Not applicable.

**Conflicts of Interest:** The authors declare no conflict of interest.

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
