# Peer review of "Antioxidant Capacity Changes and Untargeted Metabolite Profile of Broccoli during Lactic Acid Bacteria Fermentation"

_fermentation, doi:10.3390/fermentation9050474_

Round 1

Reviewer 1 Report

1. Please descript the KEGG pathways analysis method in Materials and Methods

2. Line 215 Why choose glucono delta-lactoneas as an acidifier? Why not choose lactic acid might be similar to lactic acid bacterial fermentation.

3. In section 3.5, please explain in more detail. What is the meaning if only 1-2 compounds were identified in each pathway as shown in Figure 5?

Author Response

RESPONDS TO REVIEWER #1 COMMENTS: Thanks to the reviewers for their suggestions to improve the manuscript. The RESPONSE to the comments and suggestions are in the following. 1. Please descript the KEGG pathways analysis method in Materials and Methods. Response: Thanks for the reviewer’s comments. The KEGG pathways analysis method have been added in the revised manuscript. (Section 2.10, Line 200-207) 2. Line 215 Why choose glucono delta-lactone as an acidifier? Why not choose lactic acid might be similar to lactic acid bacterial fermentation. Response: Glucono delta-lactone is a Generally Recognized as Safe (GRAS) substance and has been widely used in acidified food products to improve product texture, so we used it as an acidifier in this work. We have supplement the explain and reference in the revised manuscript (Line 274-276). Very thanks for the constructive suggestion, it’s a good idea to choose lactic acid as an acidifier to compare with the fermentation group, and we will use it in our future work. 3. In section 3.5, please explain in more detail. What is the meaning if only 1-2 compounds were identified in each pathway as shown in Figure 5? Response: Very thanks the reviewer’s comments. We have added more detail in this part in the revised manuscript. The number of compounds in each pathway means that the number of significantly differential metabolites related to the pathway screened in Section 3.4 OPLS-DA analysis. Figure 5 showed the significantly differential metabolites enriched in KEEG pathway between two groups, while not the identified compounds. If only 1-2 compounds were showed in each pathway means that compounds involved in the pathway were more similar between the two groups. (Line 352-359)

Reviewer 2 Report

* Many papers have shown that fermentation leads to an increase in the antioxidant activity of plant material, the authors should precise the novelty of their paper. Changing the plant material is not a novelty.

* Information about strain identification should be added to the materials and methods section.

* "Local market" should be changed by the name of the city everywhere in the manuscript.

* Only a few references were added in the results and discussion section. A comparison with other works should be done.

* Results about mass spectrometry are not convincing. More details about the method used should be added to the materials and methods section. How the authors grouped the molecules, and what are the molecules identified for each group in Figure 3.

* English language should be well revised

Moderate editing of English language

Author Response

RESPONDS TO REVIEWER #2 COMMENTS: Thanks to the reviewers for their suggestions to improve the manuscript. The RESPONSE to the comments and suggestions are in the following. 1. Many papers have shown that fermentation leads to an increase in the antioxidant activity of plant material, the authors should precise the novelty of their paper. Changing the plant material is not a novelty. Response: Thanks for the reviewer’s comments. A large number of LAB exist in fermented vegetables, and some strains with the ability to metabolize polyphenols are screened, but the specific transformation mechanism is not completely clear, and it can be explored in depth, to convert polyphenolic substances into substances with higher bioavailability and bioactivity through fermentation. Metabolomics approaches could help us to figure out the different expression trend of each metabolite during fermentation, which help us to understand the relevant pathways on the changed biological activity of broccoli before and after fermentation. The study aimed to provide novel insights into the quality changes in broccoli due to LAB fermentation. We have rewritten this part to precise the novelty in the revised manuscript. (Line 20-23, 48-56, 80-83, 91-93) 2. Information about strain identification should be added to the materials and methods section. Response: Thanks for the reviewer’s suggest. Information about identification of strains has added in Section 2.2. (Line 119-126) 3. "Local market" should be changed by the name of the city everywhere in the manuscript. Response: Thanks for the reviewer’s suggest. The city’s name has added in the revised manuscript. (Line 109, 119, 113-115) 4. Only a few references were added in the results and discussion section. A comparison with other works should be done. Response: Thanks for the reviewer’s comments. More discussion has been supplemented, and comparisons of results with other works have been added in the revised manuscript. (Line 247, 254-255, 284-286, 301-310, 251, 257, 286) 5. Results about mass spectrometry are not convincing. More details about the method used should be added to the materials and methods section. How the authors grouped the molecules, and what are the molecules identified for each group in Figure 3. Response: Very thanks for the constructive suggestion. There is a clear and confidence level of the identification of metabolites. As early as 2007, MSI (http://msi-workgroups.sourceforge.net), the compound appraisal group (Chemical Analysis Working Group) defines 4 confidence levels of the results of metabolites identification (Dunn et al., 2012). 2017, in the annual meeting of Metabolomics Society, a new confidence level of metabolite identification was redefined, and "level 0" was added, from 4 to 5 levels (Blazenovic et al., 2018). As shown in picture below, the higher the number, the lower the confidence level (Blazenovic et al., 2018). In our study, the In-house database (Shanghai Applied Protein Technology) of Plant Metabolome Database (Yu et al., 2022, Zhaobing et al., 2018) was used to identify the metabolites of broccoli sample. Compound identification of metabolites in broccoli samples were matched with the molecular mass of metabolites in the database (the molecular mass error was < 10 ppm), secondary fragmentation spectra, retention time and other information. MS/MS spectra with an in-house database established with available authentic standards. Confidence level is at Level 1 and Level 2. The data of molecules which have been identified in each group in Figure 3 is attached as a supplement data (Table S1) in revised manuscript. Method about metabolites annotation has been added in Section 2.10 in revised manuscript. (Line 200-207) References: Blazenovic, I., Kind, T., Ji, J., and Fiehn, O. (2018). Software Tools and Approaches for Compound Identification of LC-MS/MS Data in Metabolomics. Metabolites 8, 31. Dunn, W.B., Erban, A., Weber, R.J.M., Creek, D.J., Brown, M., Breitling, R., Hankemeier, T., Goodacre, R., Neumann, S., Kopka, J., et al. (2012). Mass appeal: metabolite identification in mass spectrometry-focused untargeted metabolomics. Metabolomics 9, 44-66.

Reviewer 3 Report

There are some issues that authors have to adress before further manuscript processing:

1. Lines 9-20 Please underline the novelty of the obtained results

2. Lines 51-63 As mentioned above. Lack of precise statment what is novel in the presented study.

3. Lines 73-82 Is this an original method? If not please add appropritae reference.

4. Lines 164-169 Here we have contradictory observations. In the first sentance authors stated that TPC did not increased significantly after 48h. And then concluded that the highest TPC was detected after 60h. To clarify please reformulate sengtences to be more precise. 

5. Lines 182-183 What is the difference between raw and broccoli fermented for 0 hours. This information is missing in methodology section.

6. Lines 192-194 Bad sentence contstruction. Please reformulate.

7.  Lines 213-214. Why those 3 time points? Why authors did not included 60h? This information should be included in the methodology section.

8. LOines 228-244 Too general. Results must be commented.

9. Lines 311-312 Please expand the thought.

4. 

English do not require much of improvement.

Author Response

RESPONDS TO REVIEWER #3 COMMENTS: Thanks to the reviewers for their suggestions to improve the manuscript. The RESPONSE to the comments and suggestions are in the following. 1. Lines 9-20 Please underline the novelty of the obtained results. Response: Thanks for the reviewer’s comments. As according to the suggestions, we have supplemented more novelty of the obtained results in Abstract. (Line 10-11, 20-23) 2. Lines 51-63 As mentioned above. Lack of precise statement what is novel in the presented study. Response: Thanks for the reviewer’s suggest. We have rewritten this part to supplement novel statement in the revised manuscript. (Line 91-96) 3. Lines 73-82 Is this an original method? If not please add appropriate reference. Response: Thanks for the reviewer’s suggest. Isolation and identification of lactic acid bacteria and other bacteria from vegetables or fruits is a routine method, the reference has been added in the revised manuscript. (Line 112) 4. Lines 164-169 Here we have contradictory observations. In the first sentence authors stated that TPC did not increased significantly after 48h. And then concluded that the highest TPC was detected after 60h. To clarify please reformulate sentences to be more precise. Response: Sorry for the unclear expression. The content of TPC was increased significantly after fermentation on each time point compare to raw broccoli, and the highest TPC was detected after 60 h’s fermentation of broccoli. But there was not significantly changed between 48 h and 60 h. We have rewritten this part in the revised manuscript. (Line 221-223) 5. Lines 182-183 What is the difference between raw and broccoli fermented for 0 hours. This information is missing in methodology section. Response: Sorry for the unclear expression. Compared with the raw broccoli puree, 0 h of fermentation time means LAB strains (LAB1, LAB10) were added into broccoli puree but without fermentation, after adding of LAB strains, the samples were immediately been stored. We have rewritten this part in the revised manuscript. (Line 139-142) 6. Lines 192-194 Bad sentence construction. Please reformulate. Response: Thanks for the reviewer’s suggest. The sentences have been modified in the revised manuscript. (Line 249-252) 7. Lines 213-214. Why those 3 time points? Why authors did not include 60h? This information should be included in the methodology section. Response: Thanks for the reviewer’s suggest. For the untargeted metabolite profile analysis, we only selected the two most representative time point, 24 h and 48 h, based on the significant variability in their TPC contents. According to the data of TPC, there was not significantly changed between 48 h and 60 h, so we did not include 60 h. The information has been added in the revised manuscript. (Line 272-274). 8. Lines 228-244 Too general. Results must be commented. Response: Thanks for the reviewer’s suggest. More comments and discussions were added in Section 3.3, and we have rewritten this part in the revised manuscript. (Line 301-310). 9. Lines 311-312 Please expand the thought. Response: Thanks for the reviewer’s comments. We have rewritten this part in the revised manuscript. (Line 392-397)

Round 2

Reviewer 2 Report

The authors addressed the comments and improved the quality of the paper.

Minor editing of English language required